# Fungal-Induced Programmed Cell Death

**DOI:** 10.3390/jof7030231

**Published:** 2021-03-20

**Authors:** Thomas J. Williams, Luis E. Gonzales-Huerta, Darius Armstrong-James

**Affiliations:** Medical Research Council Centre for Molecular Bacteriology and Infection, Imperial College London, 14 Armstrong Rd, South Kensington, London SW7 2DD, UK; thomas.williams18@imperial.ac.uk (T.J.W.); luis.gonzales@imperial.ac.uk (L.E.G.-H.)

**Keywords:** programmed cell death, apoptosis, pyroptosis, necroptosis, extracellular traps, pathogenic fungi

## Abstract

Fungal infections are a cause of morbidity in humans, and despite the availability of a range of antifungal treatments, the mortality rate remains unacceptably high. Although our knowledge of the interactions between pathogenic fungi and the host continues to grow, further research is still required to fully understand the mechanism underpinning fungal pathogenicity, which may provide new insights for the treatment of fungal disease. There is great interest regarding how microbes induce programmed cell death and what this means in terms of the immune response and resolution of infection as well as microbe-specific mechanisms that influence cell death pathways to aid in their survival and continued infection. Here, we discuss how programmed cell death is induced by fungi that commonly cause opportunistic infections, including *Candida albicans*, *Aspergillus fumigatus*, and *Cryptococcus neoformans*, the role of programmed cell death in fungal immunity, and how fungi manipulate these pathways.

## 1. Introduction

The death of the host cell during infection is a complex process which acts either as a defense mechanism or to facilitate further infection. Programmed cell deaths (PCD), both immunologically silent such as apoptosis and inflammatory such as necroptosis, can be important for the initiation of an immune response to aid in the clearance of infection. However, pathogens can also benefit from the death of the host cell as a way to remain hidden from the immune system or by being released into the extracellular space allowing for the infection of neighboring cells [1,2]. To this end, different infectious pathogens have developed a variety of mechanisms through which to influence cell death and survival pathways in order to maintain infection. Pathogenic fungi cause a broad range of infections, from superficial skin and nail infections to invasive infections affecting organs including the lungs, kidneys, and brain, with the most common opportunistic fungi, responsible for more than 90% of fungal-related deaths, being *Aspergillus*, *Candida*, *Cryptococcus*, or *Pneumocystis* species [3,4]. Those at greatest risk of invasive infections are individuals with primary immune deficiencies, such as caspase recruitment domain-containing protein 9 (CARD9) or signal transducer and activator of transcription 3 (STAT3), or acquired immune deficiencies, including those on immunosuppressants such as transplant recipients, those with cancers, and those with HIV/AIDS [5]. In spite of lower incidence than invasive bacterial infections, invasive fungal infections are of much greater concern as the associated mortality rate often exceeds 50%, killing approximately 1.5 million people each year [6]. Growing resistance to antifungals, from both clinical and agricultural applications, means that it is more important than ever that we understand how these pathogenic fungi interact with the immune system, in order to generate novel treatments for fungal diseases [7,8]. Although research into programmed cell death during infection is dominated by bacterial and viral infections, there is growing evidence that these pathways play a key role in fungal infections too. In this review, we will discuss which programmed cell death pathways are induced by pathogenic fungi, their importance in infection and how these fungi may subvert these pathways to evade the immune response.

## 2. Apoptosis

Apoptosis is a type of PCD essential for an effective immune response against pathogens and it is morphologically characterized by DNA fragmentation, organelle shrinkage, and membrane blebbing. The functional purpose of these morphological changes is to restrain any potentially harmful cytosolic content within structures called “apoptotic bodies” [9]. These apoptotic bodies are successively engulfed and cleared by macrophages, inducing an anti-inflammatory phenotype, and allowing the resolution of inflammation [10]. Apoptosis can be triggered through two different pathways: extrinsic or intrinsic. These pathways lead to the activation of caspase-3, which is considered the execution molecule for apoptosis. The extrinsic pathway is characterized by caspase-8 activation and can be mediated by stimulation of cell death receptors that bind Fas ligand (FasL), tumor necrosis factor (TNF)-α, and TNF-related apoptosis-inducing ligand (TRAIL). The intrinsic pathway is mediated by the disruption of the mitochondrial membrane and release of cytochrome C into the cytosol, with subsequent activation of caspase-3 via caspase-9. The integrity of the mitochondrial membrane depends on the balance of pro-death, Bak and Bax, and pro-survival, Mcl-1 and A1, proteins that belong to the B-cell lymphoma 2 (BCL-2) family. The delivery of pore-forming proteins, perforins, and granzymes by cytotoxic T-cells or natural killer cells also trigger apoptosis. Additionally, phagocytosis-induced cell death is triggered by production of nicotinamide adenine dinucleotide phosphate (NADPH) reactive-oxygen species (ROS) and leakage of cathepsins to the cytosol. Release of cathepsins into the cytosol may cause apoptosis through direct activation of caspase-8, or via caspase-9 activation by degrading pro-survival proteins of BCL-2 family (Figure 1) [11].

The immune response against *A. fumigatus* is mainly led by alveolar macrophages and neutrophils. In a resting state, dihydroxynaphthalene (DHN) melanin provides the characteristic grey pigment of the conidia. It has been shown in human and murine macrophages that DHN-melanin plays an important role in *A. fumigatus* virulence by activating the phosphoinositid-3 kinase (PI3K)/Akt survival signaling pathway and inhibiting extrinsic and intrinsic apoptosis pathways [12,13,14]. This mechanism enables an intracellular niche for fungal development, in which conidia increase in volume, lose the protection of their hydrophobin RodA layer, and germinate. As the hyphae grow, the secondary metabolite gliotoxin is produced, which has been shown to be an inducer of apoptosis [15]. Studies in fibroblasts and epithelial cells suggest that gliotoxin induces the activation of the c-Jun N-terminal kinase (JNK) pathway, leading to the activation of Bak through phosphorylation of Bim proteins, resulting in apoptosis [16]. Although gliotoxin induction of apoptosis has not been observed in neutrophils, the fact that Bak knockout mice have been shown to be more resistant to aspergillosis suggests that the induction of apoptosis by gliotoxin is an important mechanism for *A. fumigatus* virulence [17,18].

Phospholipomannan is a cell wall component that has been shown to induce apoptosis in J774A.1 macrophages by downregulating the phosphorylation of extracellular signal-regulated kinase (ERK)1/2-dependent Bad Ser-112 [19]. Likewise, induction of apoptosis was observed in dendritic cells treated with mannan from *Candida krusei* [20]. However, the activation of Akt signaling and subsequent inhibition of apoptosis has been reported in a different murine cell line, Ana-1, using mannoproteins of *C. albicans*. Furthermore, the dominance of anti-apoptotic signaling has been shown in proteomic analysis of RAW264.7 macrophages with *C. albicans* infection [21,22]. Further study will be required to understand how this ability to both induce and inhibit apoptosis contributes to fungal pathogenesis.

The activation of apoptosis by *Cryptococcus neoformans* is mediated by capsular polysaccharides, glucuronoxylomannan (GXM), and galactoxylomannan (GalXM). Work on murine macrophages suggested that these virulence factors mediate Fas/FasL interaction, triggering the extrinsic apoptosis pathway [23]. Later studies suggested that the intrinsic pathway is also involved in *C. neoformans* infections. Although the underlying mechanism has not been fully described, it has been shown to be dependent on nuclear factor kappa-light-chain-enhancer of activated B cells (NF-κB) signaling [24]. The effect of capsular virulence factors in the induction of apoptosis has been shown to be significant in *in vivo* models, which might suggest them to be a critical requirement for *C. neoformans* pathogenesis [25].

## 3. Pyroptosis

Pyroptosis is a form of caspase-dependent PCD and is mediated through inflammasome formation and, in comparison to apoptosis, is highly inflammatory. Inflammasome formation is initiated when intracellular sensors such as nod-like receptor (NLR) family pyrin domain containing 3 (NLRP3), absent in melanoma 2 (AIM2), or NLR family CARD domain-containing protein 4 (NLRC4) detect microbial infections or other non-infectious stimuli [26]. Apoptosis-associated speck like protein containing recruitment domain (ASC) is recruited and forms what is termed a “speck”; inactive-caspase-1 is then recruited to the ASC speck where it self-proteolytically cleaves when oligomerized, becoming active and processing pro-interleukin (IL)-1β and pro-IL-18 to their active forms [27]. Active-caspase-1 will also cleave gasdermin-D (GSDMD), allowing it to oligomerize and translocate to the cell membrane, where it forms pores that result in the swelling and eventual rupture of the cell in a highly inflammatory manner, releasing active IL-1β, IL-18, and other damage-associated molecular patterns (DAMP) (Figure 2) [28].

Phagocytosed pathogenic fungi are able to induce mature IL-1β secretion and caspase-1 activation in a two-step model. The first step is the induced expression and synthesis of IL-1β through toll like receptors (TLR) via MyD88/TRIF and through dectin-1 via the spleen tyrosine kinase-CARD9-mucosa-associated lymphoid tissue lymphoma translocation protein 1 (SYK-CARD9-MALT1) complex [29,30]. Second is the initiation of the inflammasome leading to the activation of caspase-1, which has been shown to be mediated primarily by NLRP3, but not NLRC4. Activation of the NLRP3 inflammasome has also been shown in response to fungi *in vivo* and is associated with increased secretion of IL-1β and neutrophil attracting chemokines [29,31,32,33,34,35]. ROS production and potassium efflux have been reported to contribute to the activation of the NLRP3 inflammasome in response to fungi [29,36,37]. For *A. fumigatus*, it has been shown that murine dendritic cells require both the NLRP3 and AIM2 sensors for optimal IL-1β production and antifungal defense, as mice lacking both are hypersusceptible to aspergillosis. However, mice lacking only one of these sensors exhibit little susceptibility but produce less IL-1β and IL-18, suggesting redundancy between the two sensors. It was suggested that these sensors may both recognize a common nucleic acid structure released by *A. fumigatus* that triggers the formation of a heteroduplex inflammasome [36]. Several studies have also demonstrated that inflammasome activation and IL-1β production are needed for the recruitment of other immune cells, such as neutrophils, and optimal anti-fungal activity and host defense [33,38,39].

It is suggested that although filamentation may be necessary for the induction of pyroptosis by *C. albicans*, it is not a sufficient trigger of lytic cell death, as strains that are unable to form hyphae can in fact induce pyroptosis, albeit at low levels, while some strains that have significant hyphal formation do not always induce lytic cell death [31,34]. It is suggested that the changes in the fungal cell wall act as key triggers of pyroptosis [40,41]. In particular, *C. albicans* in the germ tube stage was shown to induce the secretion of greater amounts of IL-1β than in its hyphal or yeast phase [33,34,42]. This is supported by the lower levels of cell death and IL-1β secretion induced by *C. albicans* that have deficient/delayed morphogenesis [41].

It is also suggested that ergosterol, a sterol similar to mammalian cholesterol which is a well characterized inducer of the NLRP3 inflammasome, on the fungal cell surface plays a role in the induction of pyroptosis. It is thought that during filamentation within the phagosome cell surface, ergosterol is increased in the developing hyphae, which then induces inflammasome activation [43]. This idea is supported by the direct correlation between Upc2 activity, a regulator of ergosterol levels, and the induction of pyroptosis by *C. albicans* [34]. It has also been suggested that *C. albicans* remodels its cell surface upon phagocytosis, increasing the expression of mannosylated cell surface proteins, which trigger the host response and induce cell lysis [31]. The Hog1 mitogen-activated protein kinase (MAPK) signaling cascade in *C. albicans* is central to the remodeling of the cell wall in response to the phagosome environment, as it has been shown that strains deficient in Hog1, and therefore cell wall remodeling, do not induce pyroptosis [40]. Other factors that have been suggested to contribute to the activation of the inflammasome and the induction of pyroptosis include the ability of *C. albicans* to alkalinize the phagosome environment through the production of ammonia as well as the production of the toxin candidalysin [37,44,45]. It has recently been shown that *C. albicans* can induce macrophage cell death in a two-phase model, where pyroptosis is responsible for low levels of cell death at early time points, which is later followed by glucose starvation, inducing greater levels of cell death independent of pyroptosis, although this starvation does induce the activation of the NLRP3 inflammasome. The ability to induce pyroptosis varied depending on the clinical isolate used, whereas the induction of death via glucose starvation was common to all clinical isolates [46]. However, whether the host-pathogen competition for glucose induces another form of PCD is yet to be determined.

Similarly, induction of the inflammasome and pyroptosis have also been shown to be state-dependent for *A. fumigatus* and *C. neoformans*. Specifically, *A. fumigatus* hyphae, but not conidia, and *C. neoformans* biofilms, but not yeast forms, induced inflammasome activation [29,39,47]. It is suggested that the distinct polysaccharide capsule of *C. neoformans* prevents recognition by the NLRP3 inflammasome as acapsular strains of *C. neoformans* do in fact activate the NLRP3 inflammasome [35]. It can be assumed that the protective hydrophobin RodA layer on *A. fumigatus* plays a similar role. However, it has been shown that phagocytosis of antibody- or complement-opsonized encapsulated *C. neoformans* leads to the induction of pyroptosis, thought to be the result of the release of virulence factors by *C. neoformans* which destabilize the phagolysosome [47,48].

## 4. Necroptosis

Necroptosis is an inflammatory form of PCD, which, unlike apoptosis and pyroptosis, is not caspase-dependent but receptor-interacting serine/threonine-protein kinase (RIPK)-dependent [49,50]. It was first shown to be TNFα-dependent in the presence of the pan-caspase inhibitor z-VAD-FMK (zVAD) [51], where signaling through TNF receptor 1 (TNFR1) in the absence of caspase 8 leads to the formation of the necrosome, in which RIPK1 phosphorylates RIPK3, which in turn phosphorylates the pseudokinase mixed-lineage kinase domain-like (MLKL). Once MLKL has been phosphorylated, it will translocate to the cell membrane forming pores, causing a loss of cell membrane integrity and the release of many inflammatory DAMPs [50]. Necroptosis has also been shown to be mediated via a number of other receptors, including the death receptors Fas and TRAIL, TLRs, and most recently dectin-1 (Figure 3) [52,53].

Currently, very little is known on the role of necroptosis in fungal infections and how fungi can induce or modulate this form of PCD. Recent work by Cao et al. [52] has demonstrated that fungal pathogen associated molecular patterns (PAMP) can directly induce necroptotic death in myeloid cells through dectin-1 signaling. The authors suggest that upon activation of dectin-1 the downstream signaling adaptor CARD9 directly interacts with RIPK1, facilitating the phosphorylation of RIPK3 and MLKL and the eventual cell rupture. The study shows that necroptosis is important for protecting against *C. albicans* infection *in vivo* even in the presence of caspase-8. Whether this dectin-1-CARD9-RIPK-MLKL signaling pathway is dependent on caspase deficiency is yet to be established. However, the fact that necroptosis is directly inducible from the main fungal PAMP receptor suggests that when caspases are deficient, fungal infections may induce excessive necroptotic death in phagocytes, leading to increased inflammation. Moreover, it has recently been shown that necroptosis may play an important role in allergic fungal disease. The release of bio-active IL-33 in an *Aspergillus* extract-induced murine asthma model was dependent on necroptosis, as pathology was attenuated by a necroptosis inhibitor, suggesting cell death inhibitors may be potential therapeutics in allergic disease [54].

Necroptosis has also been shown to have a role in cell–cell co-operation for controlling *A. fumigatus* fungal infection. Specifically, when germinating *A. fumigatus* conidia induce necroptosis in macrophages, an event termed metaforosis occurs, in which the germinating conidia is transferred, in a calcineurin-dependent manner, from the macrophage undergoing necroptotic death to a healthy recipient, which can then control germination [54,55]. A similar event has also been described for *C. neoformans*, termed dragocytosis; however, a cell death component for this mechanism has not yet been assessed [56]. However, if they were found to also be death-dependent, this would highlight an important role of cell death pathways in the control of fungal infections. It should be noted that in other transfer events, such as shuttling, the transfer of *A. fumigatus* conidia from a neutrophil to a macrophage, have been reported to not be dependent on death [57]. Therefore, further research is needed into when and why death-dependent fungal transfer is used in place of healthy transfer, or vice-versa.

## 5. NETosis

Neutrophils respond to a wide range of stimuli via the release of extracellular traps (ET), and have been shown to be important for the killing and capture of pathogens [58]. These neutrophils ETs (NET) contain proteins including myeloperoxidase (MPO), neutrophil elastase (NE), and calprotectin, as well as DNA [59,60,61]. The canonical formation of NETs is mediated via the production of ROS which leads to the activation of protein arginine deiminase 4 (PAD4), leading to the citrullination of histones and chromatin decondensation [62]. After nuclear decondensation, the NETs are formed within the cell and then released upon plasma membrane rupture, resulting in cell death, allowing for the capture and killing of pathogens [63,64]. NETs can also be produced in an ROS-independent non-lytic manner, where, rather than membrane rupture occurring, decondensed nuclear chromatin is expelled from the cell alongside degranulation. The NETs then form extracellularly, leaving behind anuclear cytoplasts (Figure 4) [65].

Several studies have reported the induction of NETs in response to fungi via a number of receptors, including dectin-1, dectin-2, and complement receptor (CR)-3, in response to glucans-, mannans-, and quorum-sensing molecules [66,67,68,69,70]. It has also been shown that NET formation in response to *C. albicans* can be both dependent on and independent of the extracellular matrix cell adhesion protein fibronectin, while for *A. fumigatus*, neutrophil adhesion to fibronectin is dispensable [66,69,70,71,72]. Of note, although fungi were reported to be able to induce PAD4 citrullination, it is dispensable in the process of fungal-induced NET formation [70,73]. Similarly to the PAD4 citrullination, it has been shown that NETs produced in response to fungi can be both ROS-dependent and -independent [67,70,73,74]. These studies highlight the diverse number of ways in which fungi can induce NETosis.

NETs have been shown to be released in response to *C. albicans* and *A. fumigatus*, or β-glucans alone [75,76]. These NETs have been appointed with various levels of fungicidal activity, as they are able to kill *C. albicans* in both its yeast and hyphal forms and were reported to be indispensable in the control of infections [75]. An important NET component is the antifungal protein calprotectin, a heterodimer of S100A8 and A9, shown to reduce *C. albicans* growth. Importantly, mice deficient in calprotectin develop more severe symptoms upon *C. albicans* infection [59]. In contrast, NETs released in response to A. fumigatus were found to be dispensable in the killing of the fungus, but it is suggested that calprotectin inhibits *A. fumigatus* hyphal growth in a NETosis-dependent manner [76,77].

As NETs can be potent killers of different fungal species, it is unsurprising that fungi have a number of ways to avoid their induction and reduce their efficiency. For example, it has been shown that *C. albicans* biofilms are able to inhibit NET formation, but the planktonic form does not [78]. It is suggested that this ability of *Candida* biofilms to inhibit NETs is conserved across *Candida* species as *Candida glabrata* biofilms also inhibit NET formation, albeit to a lesser degree than *C. albicans* [79]. Therefore, it can be inferred that these biofilms likely contribute to enhanced immune evasion and increased survival in response to NETs. *A. fumigatus’* hydrophobin RodA layer also gives a degree of protection from NETs, as it prevents recognition by immune cells pattern recognition receptors, reducing the number of NETs formed [80]. *A. fumigatus* has also been shown to be less susceptible to NETs compared to *Aspergillus nidulans*, which was proposed to be due to the greater N-acetyl-galactosamine (GalNAc) content of cell wall-associated galactosaminogalactan (GAG) in *A. fumigatus*. Of note, the same study found that increasing the GalNAc content of GAG in *A. nidulans* increased its resistance to NET-dependent killing [81]. *C. neoformans* appears to have a strategy to avoid inducing NETs in the form of its polysaccharide capsule, which contains GXM. In the absence of this capsule, human neutrophils were able to form NETs in an ROS- and PAD4-dependent manner, and these NETs were capable of killing fungal cells. Furthermore, it was shown that GXM blocks PMA-induced NET formation, suggesting that it protects *C. neoformans*, allowing the fungus to escape the fungicidal properties of neutrophils [82].

The release of NETs can cause excessive inflammation, which can contribute to lung injury. As NETs are released in response to *A. fumigatus* but do not contribute to the control of the infection, it is suggested that therapies to inhibit NET formation may reduce excessive inflammation in the lung and reduce lung injury. However, care should be taken in these approaches as it is thought that NETs may aid in the prevention of fungal dissemination [83,84].

## 6. Other Extracellular Traps

Although there is much research being carried out to determine the role of NETs in infection and disease, it is often overlooked that other myeloid cells are also able to form ETs. For example, macrophage ETs are formed in response to a number of different pathogens including *Staphylococcus aureus*, *Escherichia coli*, and *C. albicans*, and, similar to NETs, are associated with the death of the cell [85,86,87]. Liu et al. [88] showed that murine macrophages are able to release macrophage extracellular trap-like structures (METs-LS), containing histones, MPO, and lysozymes, in response to *C. albicans* in an ROS-independent manner. It was shown that both the yeast and hyphal forms of *C. albicans* induce METs-LS with the hyphal form being a more potent inducer. It is proposed that unlike NETs, the role of these METs-LS is not to kill but to restrict movement of the fungi, giving time for more effector cells to be recruited and preventing dissemination. J774A.1, peritoneal macrophages, and bone marrow-derived macrophages are all able to produce macrophage ETs in response to *C. albicans* which trap and kill fungal cells. However, it was also shown that the yeast may have the ability to combat these METs through the degradation of DNA [89]. It has also been reported that monocytes, prior to becoming macrophages, release extracellular DNA in response to *C. albicans*, resulting in monocyte ETs (MoET) which trap and attack the fungus. These MoETs were shown to contain many similar components to NETs, such as citrullinated H3, elastase, and MPO, which have been shown to contribute to antifungal activity [90].

There is also evidence of eosinophils releasing extracellular DNA traps along with eosinophil granules in a NADPH oxidase-dependent manner in response to various stimuli, such as immobilized immunoglobulins and calcium ionophores. These are termed eosinophil ETs (EET) and the process of their release is EETosis [91]. In regard to fungi, EETs have only been shown in response to *A. fumigatus* thus far, having been found in bronchial mucus plugs from patients with allergic bronchopulmonary aspergillosis (ABPA) [92,93]. The release of EETs in response to *A. fumigatus* was also reported in vitro, in an ROS-independent and CD11b- and Syk-dependent manner, although the traps were not able to inhibit fungal growth or have the capacity to kill the fungus [92]. This early work showing that ETs form in response to pathogenic fungi from a number of different immune cells will need to be further evaluated to elucidate their role and importance in fungal infections.

## 7. PANoptosis

Emerging data have suggested the existence of a programmed cell death pathway termed PANoptosis (Pyroptosis, Apoptosis, and Necroptosis), in which a singular complex, containing caspase-1, caspase-8, and RIPK3, is able to regulate whether a cell undergoes pyroptotic, apoptotic, or necroptotic death [94]. It has been shown that this pathway is activated in response to both *A. fumigatus* and *C. albicans* and is regulated by the intracellular sensor ZBP1 [95]. The existence of this molecular structure suggests a redundancy in cell death pathways, where if one becomes compromised, for instance by fungal inhibition, another is able to be driven. However, what this means for targeting cell death pathways for the treatment of fungal infection is yet unknown.

## 8. Conclusions

Pathogenic fungi are able to induce PCD by a number of pathways including apoptosis, pyroptosis, necroptosis, and ETosis. Future research will provide greater insight into which of these mechanisms are essential for protection and how pathogenic fungi may manipulate them. Both the induction of these pathways and how they may be avoided is dependent on a number of factors including fungal species and morphotype. Therefore, the screening of knock-out mutant strains from each species will be essential to identify and understand the virulence factors responsible for host cell death induction [45]. The type of host cell itself also determines which pathways are driven. The majority of research into pathogenic fungi and cell death is, unsurprisingly, focused around innate immune cells, the first line of immunological defense. However, these fungi affect a number of sites throughout the body, and understanding what cell deaths are induced in tissues such as the epithelium, endothelium, or organ specific tissues will be important in understanding the role of fungal-induced cell death in systemic infections. As mentioned previously, there is a growing need for treatments for fungal infections due to both the increased number of high-risk groups and the growing threat of anti-fungal resistance. Although not discussed in this review, there is evidence that fungal cells themselves undergo programmed cell death [96,97]. Once the interactions between the host and fungi in regard to cell death are further understood, the benefits of therapeutic approaches targeting these pathways could be assessed, as many other therapies manipulating the immune system are already being employed for fungal infections including monoclonal antibodies, cytokine therapies, and adoptive cell transfers [98,99]. Fully understanding the complex–death interaction between the host cell and fungi may open up a whole range of novel therapeutics to combat disease.

## Figures and Tables

**Figure 1 jof-07-00231-f001:**
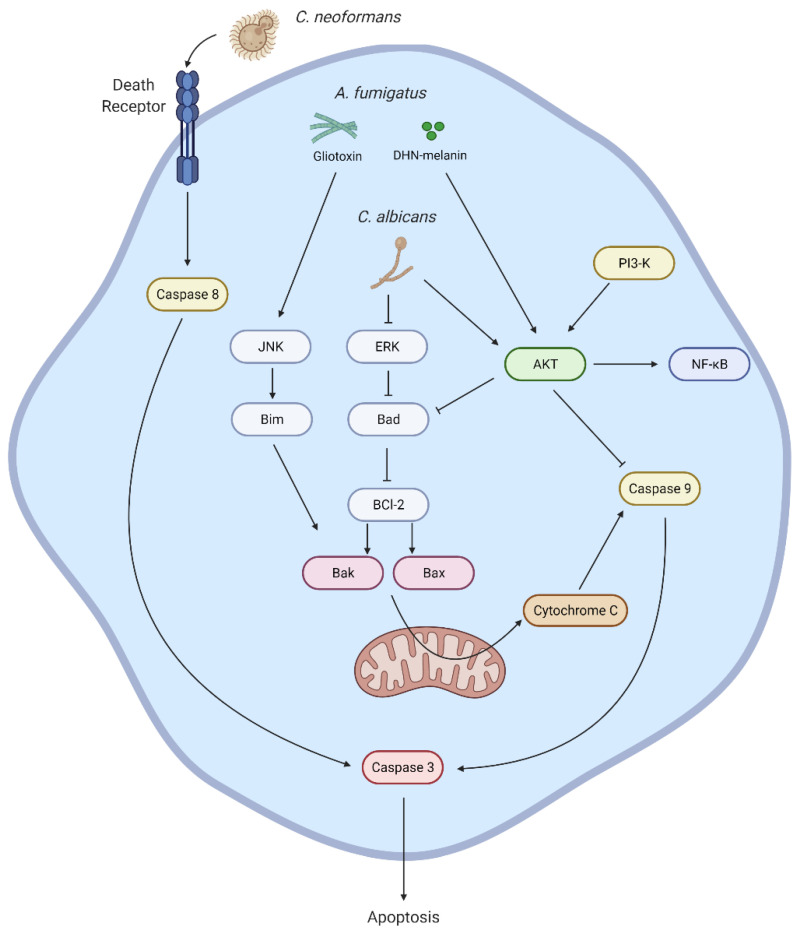
Fungal induction of apoptosis. The modulation of the extrinsic and intrinsic pathway of apoptosis depends on the type of fungal virulence factor intervening. *C. neoformans*’ capsular polysaccharides bind to cell death receptors, triggering extrinsic pathway via caspase-8 activation. Likewise, intrinsic pathway of apoptosis is induced by virulence factors in *A. fumigatus* and *Candida sp.* via modulation of B cell lymphoma 2 (BCL2) homology 3 (BH3)-only proteins Bim and Bad, respectively. Alternatively, the inhibition of apoptosis has also been described as a strategic mechanism for pathogen survival, mediated by the upregulation of Akt activity and inhibition of caspase-9.

**Figure 2 jof-07-00231-f002:**
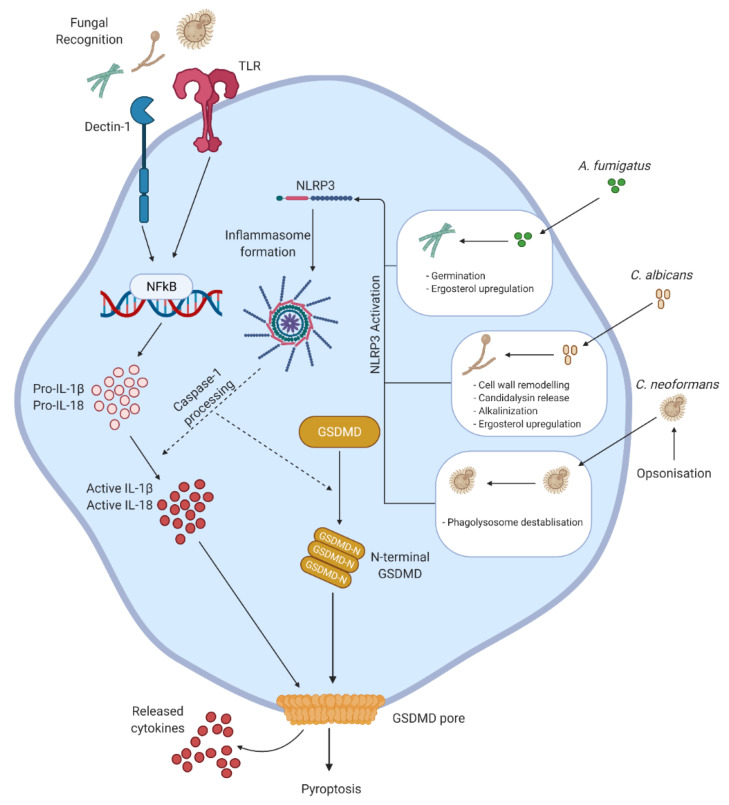
Fungal induction of pyroptosis. Recognition of fungi by patten recognition receptors such as dectin-1 and toll-like receptors (TLR) primes cells by upregulating the expression and synthesis of pro-interleukin (IL)-1β and pro-IL-18. Once the fungi are inside the phagolysosome, a number of fungal-dependent mechanisms lead to the activation of the nod-like receptor (NLR) family pyrin domain containing 3 (NLRP3), including morphological changes, ergosterol upregulation, phagolysosome alkalinization, or destabilization through the release of virulence factors. Formation of the NLRP3 inflammasome leads to the processing of caspase-1 to its active form. Active caspase-1 processes pro-IL-1β and pro-IL18 to their biologically active forms as well as cleaving gasdermin-D (GSDMD) to its N-terminal form. N-terminal GSDMD oligomerizes and translocates to the cell membrane forming pores which result in lytic cell death and the release of pro-inflammatory cytokines and damage-associated molecular patterns (DAMP).

**Figure 3 jof-07-00231-f003:**
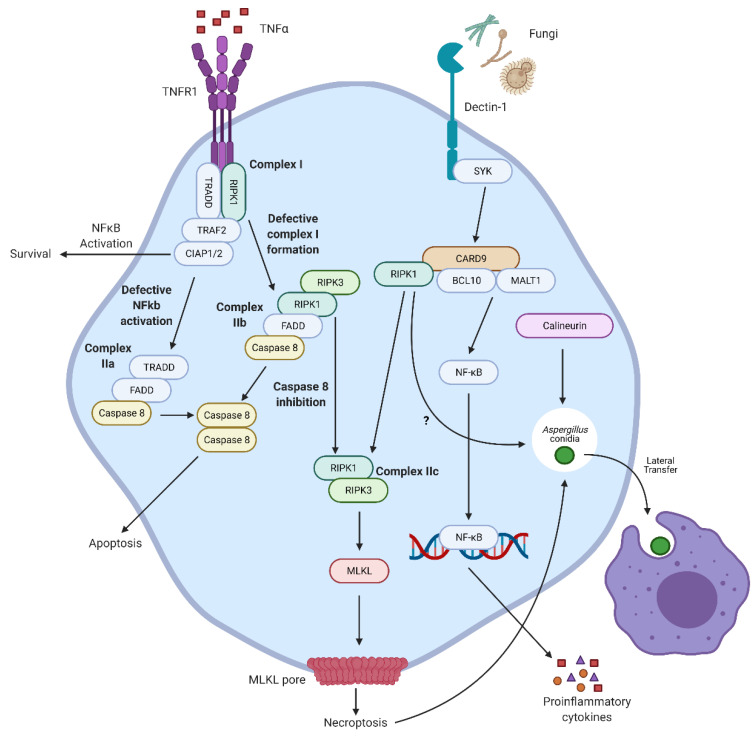
Fungal induction of necroptosis. Canonical necroptosis occurs through the recognition of tumor necrosis factor (TNF)-α by TNF receptor 1 (TNFR1). In the absence of caspase-8, receptor-interacting serine/threonine-protein kinase (RIPK)-1 is allowed to phosphorylate RIPK3 which in turn phosphorylates mixed-lineage kinase domain-like (MLKL). Phosphorylated MLKL oligomerizes and translocates to the cell membrane forming pores leading to lytic death, and therefore, the release of pro-inflammatory cytokines and damage-associated molecular patterns (DAMP). Necroptosis can be induced in response to fungi through recognition by the main fungal pattern recognition receptor dectin-1. In this pathway, caspase recruitment domain-containing protein 9 (CARD9) downstream of dectin-1 interacts with RIPK1, allowing it to phosphorylate RIPK3 and in turn the phosphorylation of MLKL, leading to necroptotic death through the specific recognition of fungi. Of note, germination of phagocytosed *A. fumigatus* conidia can lead to calcineurin-dependent lateral transfer from the infected macrophage undergoing necroptosis to a healthy macrophage to aid in fungal control, a process termed metaforosis.

**Figure 4 jof-07-00231-f004:**
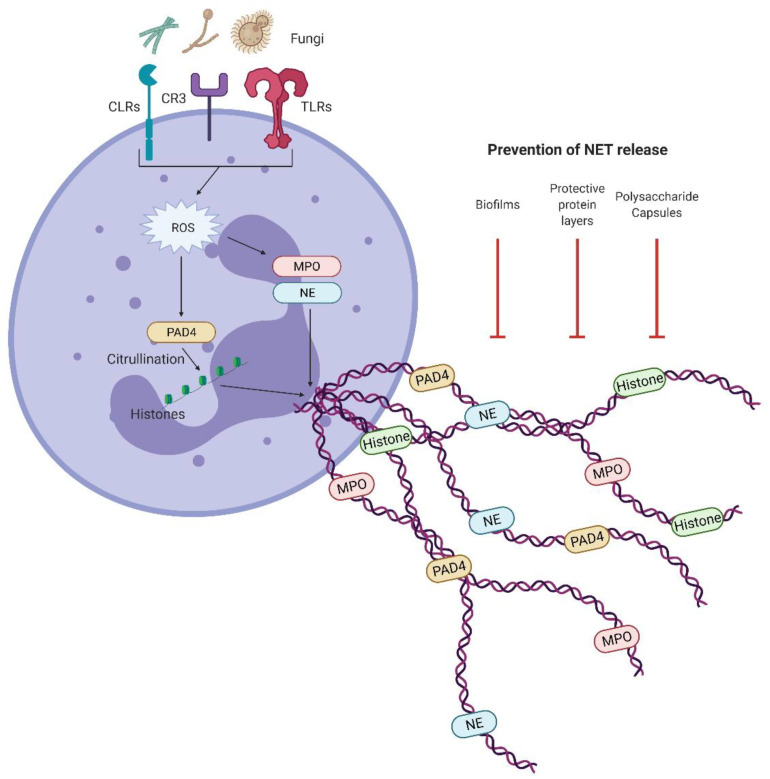
Fungal induction of NETs. Neutrophil extracellular traps (NET) are canonically formed when reactive-oxygen species (ROS) production leads to protein arginine deiminase 4 (PAD4)-dependent citrullination of histones, after which chromatin decondensation occurs and NETs are released upon cell rupture, resulting in cell death. Fungi activate a number of neutrophil receptors, including C-type lectin receptors (CLR), toll-like receptors (TLR), and complement receptor (CR)-3, to induce NETosis. NET release can occur via both ROS- and/or PAD4-dependent or -independent mechanisms. Released NETs can contain myeloperoxidase (MPO), neutrophil elastase (NE), histones, and calprotectin, and aid in antifungal activity. Fungi possess a number of features that prevent the generation of NETs including protective outer layers (*A. fumigatus* hydrophobin layer and *C. neoformans* polysaccharide capsule) or the ability to form biofilms (*C. albicans*).

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
