# Peer review of "Fungal-Induced Programmed Cell Death"

_jof, 2021, doi:10.3390/jof7030231_

Round 1

Reviewer 1 Report

This is an interesting review article that summarizes recent progress in studying how fungal pathogen stimulate different cell death pathways in host cells.  One strength of the review is that it covers a broad range of cell death pathways, including apoptosis, pyroptosis, necroptosis, NETosis, and other less well studied pathways.  Another strong point is that Figures were included to highlight the key components of these pathways.  I had only some minor comments and a few typos that need to be corrected.

  1. Paragraph starting line 145:

This section should be rewritten to include newer studies indicating that hyphae are not necessary or sufficient for pyroptosis.  The data can be found in the following papers:

Uwamahoro, mBio, 2014 PMID: 24667705

Wellington, 2012, mBio, PMID: 23269828

Wellington Uek Cell, 2014,  PMID: 24376002

O’Meara, Nat. Comm. 2015, PMID: 25824284

O’Meara mBio 2018 PMID: 30131363

  1. Related to the comments above, the paper by Tucey et al, 2020 (PMID: 32750090) should be discussed as it proposes a new model that fungi promote pyroptosis in macrophages by starving them for glucose.

  1. Line 218 describes dragocytosis as occurring in C. albicans, but the cited reference is for C. neoformans. (ref. 55)

  1. Define NE in Fig. 4.

  1. Line 321:Doesn’t the presence of MPO in the putative macrophage extracellular traps suggest they are actually looking at neutrophil extracellular traps?

Minor Comments:

Line 28: rephrase  “released to extracellularly infecting”

Line 47.  Change what to which.

Author Response

1. Paragraph starting line 145:

This section should be rewritten to include newer studies indicating that hyphae are not necessary or sufficient for pyroptosis.  The data can be found in the following papers:

Uwamahoro, mBio, 2014 PMID: 24667705

Wellington, 2012, mBio, PMID: 23269828

Wellington Uek Cell, 2014,  PMID: 24376002

O’Meara, Nat. Comm. 2015, PMID: 25824284

O’Meara mBio 2018 PMID: 30131363

            This section has been rewritten/re-arranged to include the suggested idea line 145 onwards.

2. Related to the comments above, the paper by Tucey et al, 2020 (PMID: 32750090) should be discussed as it proposes a new model that fungi promote pyroptosis in macrophages by starving them for glucose.

The idea of glucose starvation induced death and inflammasome activation has been added lines 168-176. The paper suggests that the glucose starvation activates the inflammasome but does not drive pyroptosis, see quote: “the hyphae-driven Phase I mechanism leads to pyroptotic macrophage death, but Phase II activation of the inflammasome by glucose starvation does not”

3. Line 218 describes dragocytosis as occurring in C. albicans, but the cited reference is for C. neoformans. (ref. 55)

Corrected C. albicans to C. neoformans (now line 229)

4. Define NE in Fig. 4.

NE now defined as Neutrophil Elastase in line 251.

5. Line 321:Doesn’t the presence of MPO in the putative macrophage extracellular traps suggest they are actually looking at neutrophil extracellular traps?

MPO has been shown to be found in the extracellular traps of a number of macrophage types both primary and cell lines. “MPO has been identified in the METs of diverse macrophage populations including human glomerular macrophages, human peripheral-blood monocytes, THP-1 macrophage-like cells, murine J774A.1 macrophage-like cells, bovine monocytes, and caprine monocytes”. Also the study isolates monocytes using purification by monocyte isolation kit and purity was confirmed by flow cytometry making it unlikely that neutrophils are present in the cultures.

https://doi.org/10.1159/000480373

Minor Comments:

Line 28: rephrase  “released to extracellularly infecting”

Changed to “released into the extracellular space allowing for the infection of neighboring cells”

Line 47.  Change what to which

            Changed what to which

Reviewer 2 Report

The review on "Programmed cell death induced by fungi" is well written and of interest to a wide audience; the illustrations are very clear, concise and allow an immediate understanding of the mechanisms responsible for fungal-induced programmed cell death to all types of readers. The authors carefully describe the molecular mechanisms underlying the different forms of programmed cell death (apoptosis, pyroptosis, necroptosis, NETosis, PANoptosis). They show the redundancy in the cell death pathways and their complex and often opposite role played in host protection. They also describe the multiplicity of fungal and host factors (eg fungal species and morphotype, the type of host cell) that determine the activation or inhibition of specific cell death pathways and their impact on the outcome of fungal diseases.

Author Response

No changes requested